# Integrating IoT for Soil Monitoring and Hybrid Machine Learning in Predicting Tomato Crop Disease in a Typical South India Station

**DOI:** 10.3390/s24196177

**Published:** 2024-09-24

**Authors:** Gurujukota Ramesh Babu, Mony Gokuldhev, P. S. Brahmanandam

**Affiliations:** 1Department of CSE, Vel Tech Rangarajan Dr. Sagunthala R&D Institute of Science and Technology, Chennai 600602, India; vtd991@veltech.edu.in (G.R.B.); ksmdhev@gmail.com (M.G.); 2Department of Physics & R&D Cell, Shri Vishnu Engineering College for Women (A), Bhimavaram 534202, India

**Keywords:** IoT, soil monitoring, ML algorithms, tomato crop disease prediction, soil parameter analysis, precision agriculture

## Abstract

This study develops a hybrid machine learning (ML) algorithm integrated with IoT technology to improve the accuracy and efficiency of soil monitoring and tomato crop disease prediction in Anakapalle, a south Indian station. An IoT device collected one-minute and critical soil parameters—humidity, temperature, pH values, nitrogen (N), phosphorus (P), and potassium (K), during the vegetative growth stage, which are essential for assessing soil health and optimizing crop growth. Kendall’s correlations were computed to rank these parameters for utilization in hybrid ML techniques. Various ML algorithms including K-nearest neighbors (KNN), support vector machines (SVM), decision tree (DT), random forest (RF), and logistic regression (LR) were evaluated. A novel hybrid algorithm, ‘Bayesian optimization with KNN’, was introduced to combine multiple ML techniques and enhance predictive performance. The hybrid algorithm demonstrated superior results with 95% accuracy, precision, and recall, and an F1 score of 94%, while individual ML algorithms achieved varying results: KNN (80% accuracy), SVM (82%), DT (77%), RF (80%), and LR (81%) with differing precision, recall, and F1 scores. This hybrid ML approach proved highly effective in predicting tomato crop diseases in natural environments, underscoring the synergistic benefits of IoT and advanced ML techniques in optimizing agricultural practices.

## 1. Introduction

Research in agriculture aims to enhance food productivity and quality while reducing costs and increasing profits. The agricultural production model is influenced by complex interactions involving soil, agrochemicals, and seeds. Fruits and vegetables are essential agricultural products. Plant diseases disrupt normal functions such as transpiration, photosynthesis, fertilization, pollination, and germination, affecting overall plant health. Early disease detection is, thus, crucial, requiring frequent expert monitoring, which is costly and time-consuming. Therefore, developing a rapid, cost-effective, and accurate automated disease detection technique is imperative [1].

Managing plant diseases in high-value crops like tomatoes is particularly crucial because diseases can significantly reduce both yield and quality. Tomato plants are susceptible to a variety of diseases caused by both biotic (living) and abiotic (non-living) factors [2]. Biotic factors include fungi, bacteria, viruses, and worms, while abiotic factors encompass environmental conditions such as temperature, sunlight, and nutrient availability. The presence of these factors can create an environment conducive to the proliferation of diseases, posing a constant threat to tomato crops.

Among the biotic threats, fungal diseases are particularly problematic for tomato crops. Common fungal diseases include the following:(a)**Early Blight**: Caused by the fungus *Alternaria solani*, this disease results in dark, concentric spots on leaves, stems, and fruit, leading to defoliation and reduced yield.(b)**Septoria Leaf Spot**: This disease, caused by *Septoria lycopersici*, manifests as small, water-soaked spots on leaves, which eventually turn brown and cause the leaves to wither and die.(c)**Late Blight**: Caused by *Phytophthora infestans*, late blight is infamous for its rapid spread and devastating impact on tomato crops, leading to water-soaked lesions on leaves and fruit rot.(d)**Cercospora Leaf Mold**: This disease, caused by *Cercospora* spp., produces small, circular spots with gray centers on the leaves, leading to significant leaf loss and reduced photosynthetic capacity.

Abiotic factors such as extreme temperatures, inadequate sunlight, and nutrient deficiencies can exacerbate these diseases, further stressing the plants and making them more susceptible to infections.

Disease diagnosis involves skillful, time-consuming, and error-prone visual exams. To properly address crop diseases, they must be quickly and accurately identified. Researchers have developed effective machine-learning approaches to predict tomato crop disease [3], leveraging ML’s accuracy, efficiency, and scalability. Crop visual differences can help ML approaches discover and classify disorders [4,5]. Specifying disease indicator threshold values significantly enhances the accuracy of machine learning models in predicting plant illnesses, as evidenced by recent advancements in agricultural technology. This methodology leverages both professional agronomic insights and historical data analytics, enabling a nuanced comparison that underscores the importance of integrated approaches for precise diagnosis. In particular, the severity of plant diseases, such as those affecting tomatoes, can be meticulously quantified through observable parameters like crop coloration, spot patterns, and lesion appearances, as detailed in [6]. Table 1 lists the symptoms and signs of tomato crop infection.

The efficacy of ML techniques hinges on the comprehensive collection of sensor data to train algorithms for detecting disease-specific patterns and anomalies. Among these techniques, convolutional neural networks (CNNs) are particularly noteworthy for their superior performance in image recognition tasks, offering a promising avenue for enhancing disease diagnosis accuracy [7]. This integration of machine learning with traditional agronomic knowledge heralds a new era in precision agriculture, promising significant improvements in disease management and crop yield optimization. 

Machine learning offers a superior approach to detecting diseases in tomato leaves, outperforming traditional diagnostic methods such as visual inspection or manual sample testing. By analyzing vast datasets and employing sophisticated algorithms, this technology equips farmers with powerful tools to identify and address plant diseases, thus enhancing farming efficiency, productivity, and environmental sustainability. Precision agriculture, facilitated by these advancements, optimizes resource usage and crop management [8]. It has been emphasized that the challenges faced by experts in the agricultural domain are also largely overcome by the introduction of ML techniques [9]. 

The health and yield of tomato crops are influenced by several key soil parameters, such as pH value; soil nutrients including, nitrogen (N), phosphorus (P), and potassium (K), which are collectively called NPK; and humidity, as well as weather conditions, soil temperature, availability of water, sunlight, wind, pollution level, etc. [10]. Appropriate levels of these elements are crucial, and deviations can lead to nutrient imbalances, adversely affecting crop health and predisposing the plants to various diseases. It has also been stressed that more locally specific models can produce better prediction accuracy depending on the target soil property [11]. Furthermore, though a larger dataset generally leads to more accurate soil characterization, there is no strict rule on the required size. While prediction accuracy typically improves with more samples [12], this effect can vary significantly depending on the machine learning method used [13]. 

Machine learning models utilize soil parameters, among others, to predict disease outbreaks by establishing risk thresholds and identifying data patterns indicative of potential health issues [8,14]. For instance, a detailed analysis [14] has provided a comprehensive table that correlates specific soil conditions—highlighting the optimal and critical threshold values for pH, NPK, humidity, and water content—with the likelihood of disease occurrence in tomato plants. This evidence-based approach enables targeted early intervention strategies, paving the way for more effective disease management and ultimately securing crop health and yield.

Soil parameters like pH, nutrient content, and moisture are crucial for healthy tomato cultivation and disease prevention. Tomatoes thrive in slightly acidic to neutral soil (pH 6.0–6.8). Extremes in pH cause nutrient imbalances—acidic soils lead to deficiencies in calcium and magnesium, while alkaline soils cause iron and zinc deficiencies—weakening plants and increasing susceptibility to early blight. Balanced fertilization and careful pH adjustments with organic matter or soil amendments help maintain optimal conditions and enhance disease resistance. 

Excess nitrogen can lead to overly lush vegetation, making plants more susceptible to late blight. Potassium and phosphorus deficiencies increase the risk of Cercospora crop mold [15]. Humidity and soil moisture levels are critical; overwatering or poor drainage create environments that favor late blight and crop mold. Imbalanced soil moisture can harm plants, so it is important to manage it carefully. These specific connections and threshold values are crucial for predicting potential disease outbreaks using machine learning techniques. Continuous monitoring of soil conditions using these techniques can help farmers anticipate emerging risks and take preventive actions to avoid diseases. This approach improves productivity, reduces the need for pesticides, and enhances overall plant health. 

Using real-time data and machine learning to optimize agricultural methods, this current study establishes a clear correlation between plant illnesses and soil conditions. By combining IoT and ML technology to create a reliable and effective system for soil monitoring and tomato crop disease prediction, precision agriculture may be advanced in this way. This current paper is structured as follows: A review of the literature is provided in Section 2, and the goals of the suggested study are discussed in Section 3. The study methodology is then presented in Section 4, which comes after the findings and analysis in Section 5. The results of this study are included in the conclusions and are given in Section 6, where we also discuss the present research’s future scope. 

## 2. Literature Survey

The agricultural sector has long faced challenges in implementing precision farming techniques due to infrastructural and resource constraints. However, the emergence of cost-effective internet of things (IoT) solutions has paved the way for data-driven decision-making in agriculture. IoT systems revolutionize agriculture by leveraging smart sensors to enhance farming practices. These sensors continuously monitor soil conditions, including moisture levels, temperature, and nutrient content, providing real-time data that help farmers make informed decisions. Additionally, IoT systems track plant growth by analyzing key factors such as light exposure, humidity, and growth rates, enabling precise interventions to optimize yield. One of the most significant advantages of IoT in agriculture is its ability to accurately predict crop diseases. 

By detecting subtle changes in the environment and plant health, IoT systems can forecast potential outbreaks, allowing for early intervention and reducing the risk of widespread damage. This has made precision agriculture a key approach in modern farming, with IoT technology significantly boosting crop management and productivity [16]. IoT systems incorporating environmental sensors and high-resolution cameras also capture real-time data on parameters like temperature, humidity, and soil pH [17,18].

Alongside IoT-based soil monitoring, the integration of machine learning algorithms has become crucial for the early detection and prediction of crop diseases. Several studies have demonstrated the effectiveness of various ML approaches, including support vector machines, artificial neural networks, and convolutional neural networks, in identifying plant diseases through visual analysis of plant symptoms. There is a plethora of research studies that highlight the integration of IoT and machine learning for the early detection and prediction of tomato crop diseases. Researchers use machine learning techniques, specifically convolutional neural networks (CNNs), to identify diseases like early blight, late blight, and bacterial leaf spot [19]. These systems can achieve high accuracy in disease detection. The application of deep learning techniques, notably CNNs, represents a paradigm shift in disease prediction accuracy. By analyzing vast datasets comprising soil profiles and high-resolution crop imagery, CNNs have facilitated more precise and early detection of diseases, thereby enabling proactive disease management strategies [13]. This technological breakthrough has not only improved diagnostic accuracy but also streamlined agricultural practices by optimizing resource allocation and reducing reliance on chemical interventions. 

A recent study reported not only a 95.6% accuracy using a 3-stage stacked deep convolutional autoencoder, but also that it effectively reduced the computational complexity [19]. The integration of IoT- and AI-enabled real-time monitoring and early warning systems for farmers can potentially reduce crop losses and improve overall agricultural management [20]. This approach shows promise in enhancing food security and supporting sustainable farming practices. 

The journey of agricultural discovery commenced with detailed investigations into the intricate relationships between soil nutrient variations and their impact on plant health and disease susceptibility. Early research laid foundational insights into how fluctuations in nitrogen, phosphorus, and potassium (NPK) levels influence the incidence of diseases such as early and late blight in tomatoes [21]. These findings highlighted the critical role of balanced soil nutrition in mitigating disease risks and optimizing crop yield. One pivotal advancement was the development of predictive models that not only quantified the impact of specific NPK levels on disease likelihood but also incorporated soil pH as a determinant factor. This holistic approach underscored the nuanced interactions between soil chemistry and plant health, illustrating how subtle variations can profoundly influence disease susceptibility [22]. Adding micronutrients and trace minerals to disease prediction models has helped us to learn a lot more about the complex relationships between soil health and disease, which has made disease predictions more accurate and reliable [23,24]. This holistic approach not only enhances predictive capabilities but also underscores the critical role of balanced nutrition management in sustainable agriculture practices. 

By integrating soil health assessments with disease prediction models and advocating for sustainable farming practices, a novel approach emphasizes a reduction in pesticide dependence, thereby promoting environmental sustainability [25]. Research exploring the impact of water availability and soil humidity on NPK levels and disease occurrence has provided deeper insights into the multifaceted factors influencing plant health [26]. Such investigations are pivotal in refining agricultural practices to optimize nutrient utilization and disease management strategies.

In the realm of classification methods, support vector machines (SVMs) have demonstrated success in plant disease detection due to their ability to handle complex datasets and nonlinear relationships [27]. However, SVMs require meticulous kernel selection and parameter tuning to achieve optimal performance, which can be resource-intensive. Dense convolutional neural networks (DCNNs) and other deep learning architectures have emerged as powerful tools for high-accuracy plant disease detection [28]. While CNNs excel at extracting intricate patterns from data, their computational demands and reliance on large training datasets pose challenges in practical implementation [29,30]. These limitations underscore the ongoing need for efficient computational resources and robust data management strategies to harness the full potential of deep learning in agricultural applications. 

Machine learning models, exemplified by XGBoost, have proven effective in predicting greenhouse tomato crop evapotranspiration through the analysis of meteorological data [31]. This capability extends to disease prediction by integrating diverse environmental and soil parameters as input features, thereby enhancing the models’ predictive power across varying agricultural contexts. Furthermore, hybrid machine learning approaches integrating spectral and structural data from UAV and satellite imagery have shown significant improvements in monitoring crop health and predicting yield-related metrics, underscoring their potential for comprehensive agricultural management [32].

One notable study [33] utilized advanced AI and computer vision techniques for the early classification of tomato diseases. DenseNet121 achieved impressive results with 99.88% training and 99.00% testing accuracy, while ResNet50V2 and ViT followed closely with 95.60% and 98.00% accuracy, respectively. Despite their efficacy, these models struggle with high computational demands and extensive data preparation. Nonetheless, their diagnostic capabilities are vital for improving crop yield and quality.

In a similar vein, a research study employing CNNs achieved a remarkable 98.49% accuracy in diagnosing tomato leaf diseases using a dataset of 3000 images [34]. However, the computational intensity of CNNs poses challenges for real-time field applications, requiring continuous dataset updates to sustain accuracy against evolving disease strains. Comparative studies across varied climates highlight the critical need for region-specific models in crop disease detection [35]. Environmental factors like temperature, humidity, rainfall, and soil characteristics vary widely across regions, affecting crop health, disease prevalence, and nutritional balance. For instance, diseases common in humid, warm climates may not appear in cooler, arid regions, making localized disease detection and management crucial. Previous studies have faced challenges such as imbalanced datasets, limited real-world data, and inadequate field testing. This study is, therefore, significant as it assesses tomato diseases in a new location in South India, using IoT-based soil monitoring in real-time and hybrid machine learning models to offer valuable insights for farmers and researchers. Furthermore, Table 2 summarizes key studies on integrating IoT and machine learning for early tomato disease detection, highlighting their limitations, which also motivated our research study. 

## 3. Objectives of This Present Research

According to the literature review, earlier works have suffered from one or more shortcomings. For instance, in one study, the system’s accuracy in identifying blight disease needs improvement, and both image analysis and machine learning may suffer from issues with accuracy, robustness, and generalizability [17]. Several studies face limitations, including a lack of true real-time capabilities and limited testing on only benchmark datasets [19], requiring further research for real-world evaluation. Disease categorization by models might not be comprehensive, and systems may not consistently predict diseases accurately [20]. Sensor and algorithm accuracy and reliability could be limited [20], and hardware wear and tear necessitates regular maintenance [36]. The proposed systems are still in development and have not been fully implemented or compared to other models, leaving their real-world readiness uncertain [37]. Potential variability in CNN accuracy, sensor calibration issues, and unaccounted environmental factors could also affect system performance [38]. When integrating advanced technologies such as soil moisture sensors, environmental sensors, and machine learning models like Azure Custom Vision into agricultural practices, several potential challenges and inaccuracies may arise. These challenges could significantly impact the reliability and effectiveness of the system, especially in real-time applications [39].

Compared to previous research works, this present research offers several key advantages. Firstly, it utilizes a hybrid machine learning approach that can analyze complex data more thoroughly than traditional methods. Secondly, it addresses the scarcity of region-specific models by incorporating real-time data tailored to diverse climatic factors, such as humidity, soil types, and disease rates. This customization is crucial for developing effective models for different locations. Thirdly, it emphasizes the need for comprehensive analyses of evaluation metrics to enhance resource management.

The goals of this research are, therefore, as follows:(a)Develop a hybrid machine learning algorithm integrated with IoT to enhance the accuracy and efficiency of soil monitoring.(b)Create models tailored to specific climates and regions for accurate crop disease detection.(c)Evaluate various ML algorithms, including K-nearest neighbors (KNN), support vector machines (SVM), decision tree (DT), random forest (RF), and logistic regression (LR) for disease prediction accuracy.(d)Introduce and assess a novel hybrid algorithm called ‘Bayesian optimization with KNN’ to combine multiple ML techniques and improve predictive performance.(e)Demonstrate the hybrid algorithm’s effectiveness in predicting tomato crop diseases, achieving superior results with 95% accuracy, precision, and recall and an F1 score of 94% compared to individual ML algorithms.

## 4. Data and Methodology

### 4.1. Tomato Crop Location

This present study considered tomato fields located at Anakapalle (geographic latitude 17.6896° N, geographic longitude 83.0024° E, MSL ~ 29 m), a major municipality in the state of Andhra Pradesh, south India. Figure 1 shows the map of India and the location of Anakapalle. Tomatoes (*Solanum lycopersicum*) flourish in Andhra Pradesh’s diverse agro-climatic conditions, which span from coastal regions to fertile interior plains. The state’s strategic location and varied seasons enable year-round cultivation, ensuring a steady supply of this essential vegetable. Andhra Pradesh, the largest tomato-producing state in India, achieved a notable production of 2744.32 thousand metric tons in the last financial year, i.e., FY 2023–2024. 

This study utilized one-minute datasets gathered meticulously from fields in Anakapalle (India) over five consecutive days, from 27 November to 1 December 2023. The following lines explain the rationale for selecting only five days in this research.

Tomatoes are most susceptible to diseases during the following:(a)Seedling Stage: Prone to damping-off, causing wilting and death.(b)Vegetative Growth Stage: Vulnerable to fungal diseases like **early blight** and bacterial infections.(c)Flowering Stage: Susceptible to blossom end rot due to calcium deficiency.(d)Fruit Development Stage: Commonly affected by anthracnose and various fungal infections.

The leaf blight phase, often known as early blight, represents a critical and destructive stage in the disease progression of tomato plants. This phase is particularly notorious for its potential to cause significant damage to tomato crops, especially under conditions of severe incidence. When left unchecked, early blight can escalate to such an extent that it leads to total loss of the crop, leaving farmers with devastating consequences. The importance of managing this phase effectively cannot be overstated, as highlighted in [40]. Early blight is not just a minor affliction but is recognized as a major factor in the reduction in tomato crop yields. It attacks the foliage, stems, and fruit, weakening the plant’s overall health and productivity and the destructive nature of early blight, particularly during the leaf blight phase, makes it one of the most critical challenges faced by tomato cultivators [41]. 

Most importantly, early blight, caused by *Alternaria solani*, leads to severe defoliation of tomato crops under conditions of heavy rainfall, high humidity, and temperatures between 24 and 29 °C [41]. Yield losses due to this disease vary by region, ranging from 35 to 78% across India, the USA, Australia, Israel, and the UK [42]. This present research, therefore, focuses on the ‘Vegetative Growth Stage’ by only collecting the most important nutrient (NPK) and soil parameter data. It has been stressed that more locally specific models can produce better prediction accuracy depending on the target soil property [43]. A thorough review of early blight (EB) in tomatoes is discussed in [44]. For clarity, Figure 2 presents probability density functions (PDFs) of the datasets collected during our research. 

The provided histograms with PDFs for nitrogen (%), potassium (%), phosphorus (%), humidity (%), soil pH, and temperature (°C) all display approximately normal distributions, with most data points clustering around the mean values. Nitrogen is centered at 2%, potassium around 3%, phosphorus around 1%, humidity around 80%, soil pH around 6.5, and temperature around 25 °C, each showing a slight right skew. The close alignment of the PDFs with the histograms indicates that the datasets are normally distributed and suitable for statistical analysis and modeling.

### 4.2. Data Collection—Arduino Microcontroller

Modern farms leverage advanced sensors to decode the environmental complexities of soil and air, which are vital to crop health and yield. These engineered devices, blending metal and plastic, operate seamlessly to collect and analyze data. It is worth mentioning that the IoT systems placed in the field provided real-time farm data with time-stamped entries, offering minute-by-minute time series updates. This granularity allows for precise monitoring of field conditions. Figure 3 shows the Arduino microcontroller used to collect soil parameters, including pH value, temperature, humidity, and other relevant data in this present research.

A GPS sensor surveys the farm’s topography, ensuring uniform sunlight distribution for even growth. An RGB color sensor, Sarah’s tool, monitors soil hues to assess health and fertility, which is essential for vine sustenance [45]. An air quality sensor discreetly measures particulate matter [46], wirelessly sending data to Sarah’s mainframe to maintain pure air. Temperature and humidity sensors protect the fruits, with their crucial readings streamed to the cloud via an ESP8266 Wi-Fi microcontroller, enabling real-time data processing [47]. A dedicated NPK sensor ensures optimal nutrient levels and balanced pH for the vines, streaming data to a cloud-based platform via an Arduino board. This board acts as the farm’s AI, learning, adapting, and issuing directives based on continuous data flow and maintaining vigilant environmental monitoring. Powered by a 24 V battery system, this network of sensors and technology upholds the equilibrium for the vineyard’s flourishing, as shown in the attached diagram [48]. Table 3 depicts the various sensors used in this present study. An attempt was also made to make a comparison between soil parameters (temperature and humidity) measured by the Arduino sensor and those provided by the India Meteorological Department (IMD, https://mausam.imd.gov.in/), and almost a one-to-one similitude was found, which implies that this research used only quality data.

### 4.3. Calculation of Kendall’s Correlation (τ)

Kendall’s correlation, named after Maurice Kendall, is essential in hybrid machine learning for feature selection and model evaluation due to its robustness to outliers and ability to handle ordinal data. In agricultural data management, it analyses datasets comprising temporal markers, temperature, and humidity, pH, and NPK levels. The process begins with rigorous data preparation, addressing missing values through imputation or removal to prevent bias. Categorical and timestamp data are encoded to numerical values and used to calculate Kendall’s correlation.

By applying the Kendall’s correlation coefficient, this study seeks to establish monotonic relationships between environmental factors and the health status of the plants. The non-parametric nature of this correlation is suitable for datasets that may not adhere to a normal distribution and can handle ordinal and continuous variables alike. The processed dataset is thus poised for use in classification algorithms, such as decision trees (DT), support vector machines (SVM), or neural networks (NN), to predict disease presence in the paddy field. The goal is to leverage this model to forecast and mitigate crop diseases, thereby enhancing yield and reducing losses due to plant health issues. The dataset provides a basis for developing a predictive model that can be trained, tested, and validated for accuracy and reliability in real-world agricultural settings. 

The Kendall’s correlation coefficient (τ) is calculated as
$$\tau =\frac{2(n_{c}-n_{d})}{n(n-1)}$$
where $n_{c}$ is the number of concordant pairs, $n_{d}$ is the number of discordant pairs, and n is the total number of pairs. 

The algorithm iteratively evaluates all column pairs, identifying concordant and discordant pairs based on their relative order. Concordant pairs are those where both elements are greater or lesser than another pair, while discordant pairs have differing orders. The coefficient τ ranges from −1 (complete disagreement) to 1 (perfect agreement), with 0 indicating no association. Interpreting these coefficients reveals intervariable relationships, aiding in factor selection, trend analysis, and predictive modeling. This process unveils insights for informed decision-making and strategic planning in agricultural management.

### 4.4. Bayesian Optimization with the KNN Algorithm 

Bayesian optimization is a powerful method for optimizing performance metrics, especially in hyperparameter tuning, where each evaluation requires significant computational resources and time [49]. It is effective for finding the optimal hyperparameters for K-nearest neighbors (KNN) models by defining an objective function, such as model accuracy. The process starts with setting a prior distribution over the hyperparameters and using an acquisition function like expected improvement (EI) to balance exploration and exploitation. 

The algorithm begins with random initial hyperparameters, evaluates the objective function, and updates a surrogate model that approximates the objective function. Iteratively, it uses the surrogate model and acquisition function to select new hyperparameters, refining the model until convergence criteria or iteration limits are met. The optimal hyperparameters are then used to train the KNN model on the entire dataset, aiming to outperform models trained with randomly chosen hyperparameters. Finally, the fine-tuned KNN model is tested on an independent dataset to assess its generalization and performance, efficiently reducing the computational cost and time of exhaustive search methods.

To explain the optimization process using Bayesian optimization and the K-nearest neighbors (KNN) algorithm, the ‘divide and explain’ rule will be generally used, as explained in the following: (a)KNN is a non-parametric, instance-based learning algorithm that classifies a new data point based on the majority class among its N-nearest neighbors in the feature space.

Distance metric: the most common distance metric used in KNN is the Euclidean distance and, mathematically, it can be written as:$$d(x,\;x_{i}\sqrt{\sum_{j=1}^{n}\left(x_{j}-x_{ij}\right)}$$
where

x = (x_1_, x_2, ………_ x_n_) is the new data point;

x = (x_i1_, x_i2, ………_ x_in_) is the *i*th data point in the training set. 

Classification rule: the class of new data points is determined by the majority class among the k nearest neighbors:$$\hat{y}=mode\;\left\{y_{i1},\;y_{i2,\;\ldots ,\;}y_{ik}\right\}$$
$$where\;y_{i1},\;y_{i2,\;\ldots ,\;}y_{ik\;are\;the\;labels\;of\;the\;k\;nearest\;neighbors}.$$

(b)Bayesian Optimization:

Bayesian optimization is used to optimize hyperparameters (like the number of neighbors k in KNN) by building a probabilistic model of the objective function and iteratively selecting hyperparameters to evaluate based on this model, which includes the following:Objective function.Gaussian process regression.Acquisition function. Common choices in acquisition include the following:(i)Expected improvement (EI).(ii)Probability of improvement (PI).
(c)Combining KNN with Bayesian Optimization:
Hyperparameter *k*: in the context of KNN, the primary hyperparameter is the number of neighbors *k*.Optimization: using Bayesian optimization to find the optimal k that maximizes the classification accuracy.

Example: To optimize *k* for a KNN classifier, do the following:Define *f*(*k*) as the cross-validated accuracy of the KNN classifier.Use Bayesian optimization to find the *k* that maximizes *f*(*k*).

This approach systematically and efficiently finds the optimal hyperparameters, improving the performance of the KNN model.

### 4.5. Data Preparation and Plant Health Classification 

To leverage IoT, a complex Arduino-based circuit terminal was used to systematically collect soil samples and gather data on environmental parameters such as humidity, temperature, and nutrient content. After data collection, the data were securely transmitted to the cloud for further analysis. The first step was data cleansing, which was accurately performed using Python 3.13. This involved removing any extraneous or disruptive information to retain only the most pertinent data. Python’s capabilities were then used to optimize and synchronize the data structure. Next, Python’s analytical tools were employed for correlation research to uncover links and trends within the processed data, essential for gaining insights into environmental factors affecting crop health. 

In the predictive modeling stage, sophisticated algorithms like KNN and Bayesian approaches were applied. Python’s Scikit-learn and Matplotlib were used for comparative analysis of various prediction techniques. This research demonstrates that incorporating IoT technology into data processing and analysis can enhance agricultural operations and open new avenues in precision farming. It illustrates the potential success of integrating traditional farming with cutting-edge technology. The IoT system placed at the field provided real-time farm data with time-stamped entries, offering minute-by-minute time series updates. This granularity allows for precise monitoring of field conditions. Table 4 shows how we prepared the collected data to propose a hybrid model.

(a)The dataset was initially collected through automated sensors, recording key environmental parameters every minute. Each entry in the raw dataset included a timestamp, NPK levels, temperature in degrees Celsius, humidity percentage, and soil pH values. These measurements, critical for monitoring crop conditions, were systematically logged, reflecting the real-time state of the agricultural environment. During processing, a new “Diseased” column was introduced to the dataset. This column serves as a binary indicator, where a value of ‘1’ signifies the presence of disease in the crop at the time of data collection, and a ‘0’ indicates a healthy condition. The addition of this column allowed for the seamless integration of raw sensor data with the results of subsequent analysis, providing a clear, unified view of both the environmental conditions and their impact on crop health.(b)Consolidating raw and processed data into a single table enhances dataset efficiency for analysis. This approach ensures that all relevant information, from the initial environmental readings to the final disease classification, is presented in a cohesive manner, offering a complete picture of the dataset’s evolution from collection to its processed state.

### 4.6. Proposed Framework—Graphical Representation 

Figure 4 illustrates the proposed hybrid ML-enabled framework, supported by an IoT system for real-time soil nutrient monitoring. The high-resolution image shown in Figure 4 showcases a hybrid ML-enabled framework for real-time soil nutrient monitoring, which integrates various sensors (NPK, temperature and humidity, and GPS) connected to an analog-to-digital converter. The data from these sensors are transmitted to the cloud, where they are used for training a machine learning model that predicts crop health. 

The following lines provide the novelty aspects of this present work compared to previous research works:(a)Integrated Sensor System: Unlike previous frameworks that may use isolated sensors and limited datasets [50], this approach combines NPK, temperature and humidity, and GPS sensors to capture a comprehensive set of environmental and soil parameters in real-time.(b)Real-Time Monitoring: The integration with cloud computing allows for continuous monitoring and data processing, enabling immediate responses to changes in soil conditions. This is an improvement over past methods that might only provide periodic data updates [51].(c)Model Training with Real-Time Data: The model is continuously trained with real-time data, enhancing its accuracy and reliability over time.

This framework’s novelty, therefore, lies in its holistic, real-time approach, leveraging multiple sensors and cloud-based ML to deliver precise and timely predictions for crop health management.

## 5. Results and Discussion

This study successfully developed a hybrid ML algorithm, enhanced with IoT technology, for soil monitoring and tomato crop disease prediction in Anakapalle, a south Indian station. Key soil parameters, such as humidity, temperature, pH, nitrogen (N), phosphorus (P), and potassium (K), were monitored using an IoT device during the vegetative growth stage. The analysis of these parameters through Kendall’s correlations facilitated their ranking, which was crucial for optimizing the hybrid ML techniques. The results of this research are presented in the following lines. 

The receiver-operating characteristic (ROC) curve shown in Figure 5 is steep and reaches close to the top-left corner, which is ideal. With an area under curve (AUC) of 0.95, the classifier demonstrates excellent performance, effectively distinguishing between positive and negative classes with minimal overlap. The closer the curve is to the upper-left corner and the higher the AUC value, the better the model’s predictive capability, making this classifier highly reliable. Soil amendments with green manures like fodder radish and bio-enriched organic manures combined with macro- and micronutrients enhance soil fertility, growth, yield, and disease resistance in tomato crops [52,53]. The ROC curve likely represents the effectiveness of these nutrient management strategies in improving tomato crop health, with an AUC of 0.95 indicating high discriminative ability in distinguishing healthy from diseased crops. 

Figure 6 compares the performance metrics of a machine learning model on training and testing datasets. The minimal divergence in the results across accuracy, precision, recall, and F1 score suggests a consistent pattern, indicating that the model generalizes well to unseen data and is well-tuned, avoiding overfitting. Specifically, we expect a slight elevation in the training metrics, given that models typically outperform on data they have already trained on. 

Test metrics, on the other hand, do not lag behind, indicating that the model maintains its robust performance even in the presence of fresh data. For the model to be dependable and useful when applied in real-world situations there must be a match between the training and test results. 

The graph reflects strong overall performance, with high values in accuracy, precision, recall, and the F1 score. These metrics collectively point toward the model’s capability to correctly identify positive cases, precisely predict the positive class, and balance the trade-off between precision and recall effectively. Such a performance profile indicates the model’s suitability for practical application, although careful consideration for potential false positives and negatives remains necessary to fine-tune its predictive accuracy further. This vigilance will enhance the model’s utility and trustworthiness in practical settings, solidifying its strength as a predictive tool. 

With 5900 true positives and 315 genuine negatives, the model can accurately classify situations as healthy or unhealthy. It must be admitted that the model made 315 false positives and 670 false negatives in disease diagnosis. The statistics indicate that the classifier performs exceptionally well in identifying positive instances, as evidenced by the high number of true positives (5900). On the other hand, with 315 false positives, the classifier oftentimes incorrectly identifies negative instances as positive, and the classifier missed 670 positive instances, which could be critical depending on the application context. 

Performance metrics, shown in Figure 7, highlight similarities among various machine learning algorithms. The best algorithm is “Bayesian Optimization with KNN”, with an accuracy, precision, and recall of 0.95, and an F1 score of 0.94, making it highly adaptable. “SVM” and “Logistic Regression” also perform well, with accuracies of 0.82 and 0.81, respectively. “Logistic Regression” excels in recall at 0.94, ideal for applications prioritizing true positives. Conversely, “Decision Tree” underperforms across all metrics, facing accuracy and precision challenges. “Random Forest” achieves balanced performance, fitting all criteria. A comparison between the hybrid model (Bayesian optimization with KNN) and the other models shows how much better the hybrid model performs overall. It is expected that data scientists and practitioners may find these insights useful in choosing the right algorithms. Table 5 provides an overview of earlier studies and sets our work apart from theirs.

## 6. Conclusions

Modern agriculture faces significant challenges in soil monitoring and disease prediction for crop management. Integrating advanced machine learning algorithms with IoT technology offers a potential solution, particularly for tomato crops. Using ML techniques, this research reports that tomato crop disease detection performance can be improved and can subsequently improve farming operations. This present study proposes a hybrid ML that combines multiple ML techniques to improve predictive performance. 

To fine-tune the model’s hyperparameters, this study employed Bayesian optimization with KNN. As shown in Figure 7, the proposed optimization model surpasses existing methods, achieving high accuracy, precision, and recall, all at 95%. Its stable F1 score of 0.94 reflects strong forecasting accuracy. By utilizing IoT for real-time data collection and hybrid ML for advanced data analysis, this integrated approach aims to enhance decision-making in tomato cultivation, improving both yield quality and quantity. Additionally, this study explores tomato crop prediction in a relatively new region of South India, where such research has not been previously reported. Therefore, this research serves as a valuable case study for agricultural scientists, farmers, and other stakeholders. 

As far as the limitations of this are concerned, while the hybrid ML model shows high performance, its scalability to larger datasets and different agricultural contexts remains untested. This study focuses primarily on tomato crops, leaving the model’s effectiveness for other crops or multiple disease types unexplored. The reliance on Bayesian optimization with KNN for hyperparameter tuning, while effective, may limit broader adoption due to its complexity. Future research is needed to improve scalability, accuracy, and data integration, indicating that the current model, though promising, is not yet ready for widespread use.

In future work, we aim to extend data collection across multiple growth cycles, potentially covering an entire season or into the following year. This will provide a deeper understanding of disease progression and environmental variability, enhancing the robustness and applicability of the results while addressing concerns about dataset adequacy. 

In the future, to improve the performance of these creative works further, future research should concentrate on enhancing the scalability and accuracy of machine learning algorithms, consumer food security, and the integration of other data sources. Future research should also focus on expanding the dataset to include more diverse agricultural environments and disease types. Research efforts should focus on developing robust and scalable ML algorithms to handle complex soil nutrient data and integrate ML techniques with sensor networks and remote sensing (satellite-based) data to realize their full potential in soil science.

## Figures and Tables

**Figure 1 sensors-24-06177-f001:**
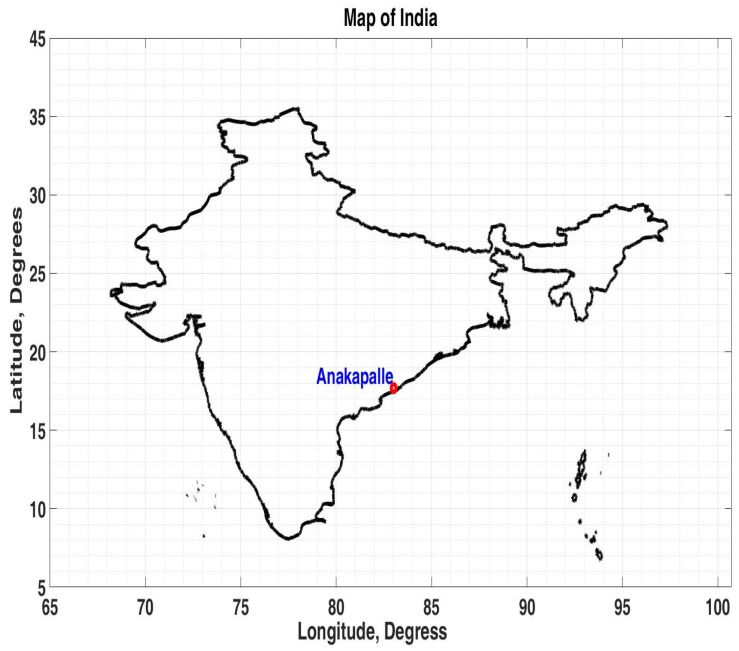
Map of India illustrating the location of Anakapalle. Natural Earth data, https://www.naturalearthdata.com (accessed on 21 July 2024), were utilized to generate this figure.

**Figure 2 sensors-24-06177-f002:**
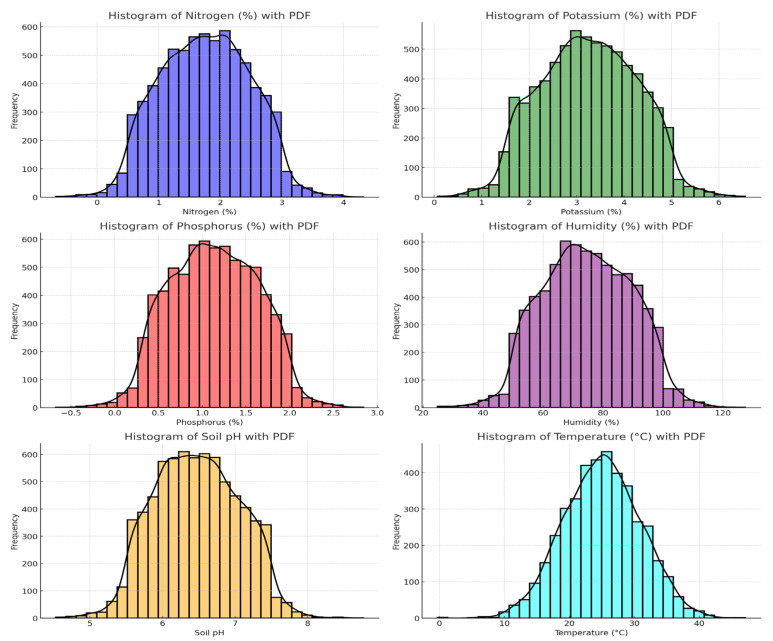
Probability density functions (PDFs) of the datasets we collected as part of this research.

**Figure 3 sensors-24-06177-f003:**
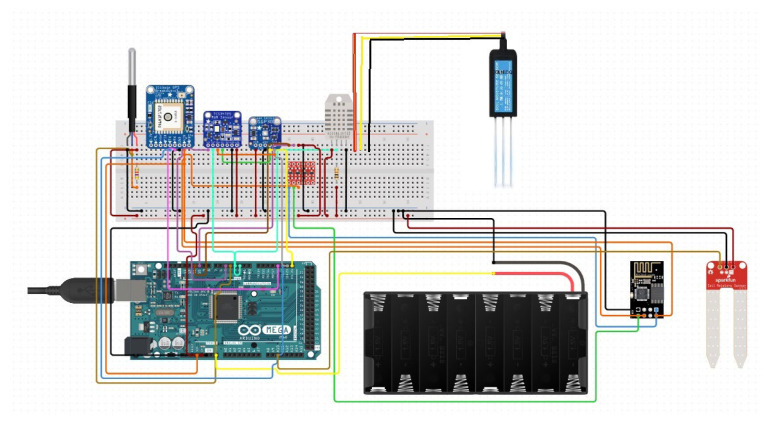
The Arduino microcontroller used in this present research.

**Figure 4 sensors-24-06177-f004:**
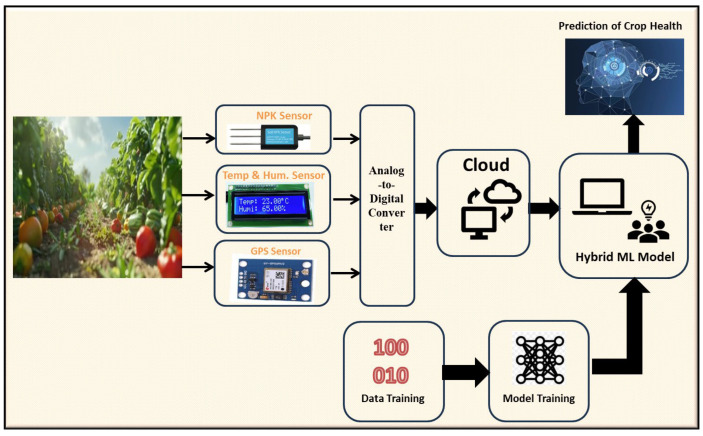
Proposed framework of this present research study.

**Figure 5 sensors-24-06177-f005:**
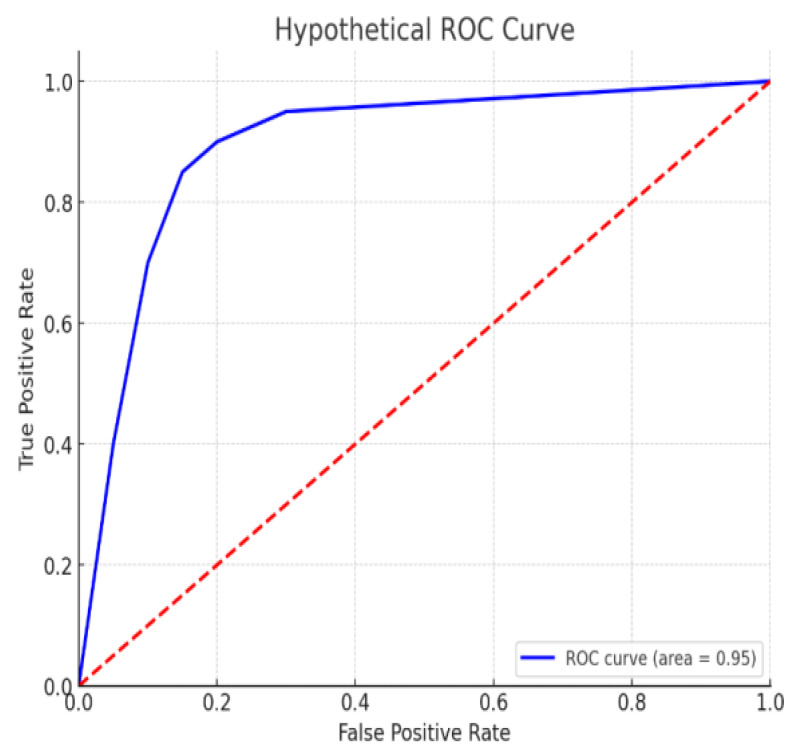
ROC curve illustrating a highly effective classifier (AUC = 0.95), wherein the red dotted line represents the random classifier performance.

**Figure 6 sensors-24-06177-f006:**
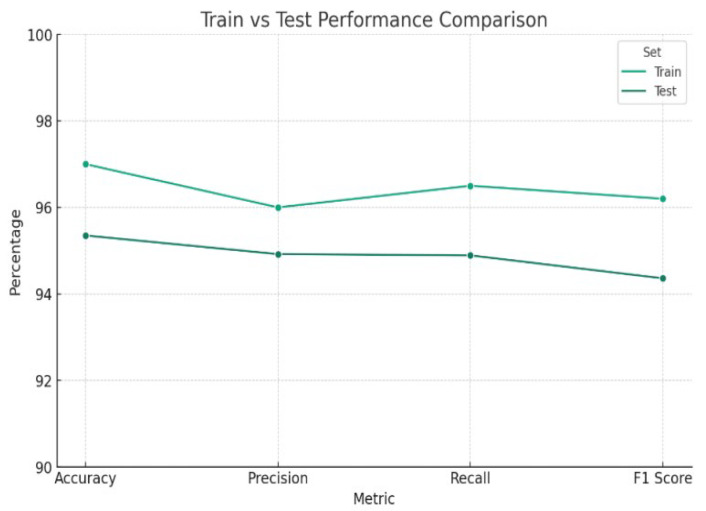
Training and test line graph.

**Figure 7 sensors-24-06177-f007:**
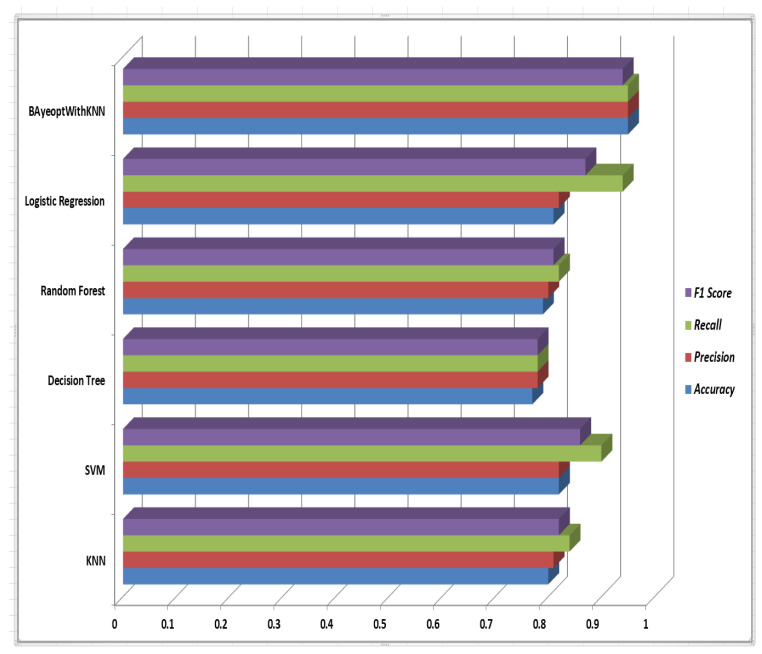
Comparison of performance metrics across ML algorithms. Our proposed method, BayeoptWithKNN, outperforms others, consistently achieving the highest performance metrics.

**Table 1 sensors-24-06177-t001:** Tomato crop disease features [2].

Indicator/Symptom	Early Blight	Late Blight	Cercospora Crop Mold
Lesion shape	Circular	Irregular	Irregular
Lesion color	Brown	Grey	Pale green
Crop spotting	Common	Common	Rare
Lesion margin	Defined	Undefined	Undefined
Sporulation	Moderate	High	Low

**Table 2 sensors-24-06177-t002:** Key studies on IoT and machine learning integration for early tomato disease detection.

S. No	Paper Title and Reference	Summarized Results	Adopted Methodology	Limitations
1.	IoT-Based Blight Severity Disease Recognition System in Tomato Plant [17]	An IoT system with sensors and machine learning models can accurately identify and categorize early and late blight diseases in tomato plants.	Monitoring environmental parameters with IoT sensors. Capturing high-resolution images of tomato leaves. Analyzing images using CNNs and deep learning. Categorizing disease severity.	The system’s accuracy in identifying blight diseases needs improvement. Image analysis and machine learning may have limitations in accuracy, robustness, and generalizability. Disease categorization by the model may not be fully comprehensive.
2.	The integration of machine learning and IoT for the early detection of tomato leaf disease in real-time [19]	The paper presents an IoT-driven system leveraging machine learning for early detection of tomato leaf diseases.	CNN extracts low- and high-level features. Integration forms a hybrid discriminative feature space. 3-stage stacked deep convolutional autoencoder reduces complexity and optimizes CNN performance. Model achieves 95.6% accuracy on Plant village dataset with 5-fold cross-validation.	The system lacks true real-time capabilities, requiring further research. The model was tested only on a benchmark dataset; more research is needed for real-world evaluation.
3.	Review on Prediction of Crop Disease Using IoT and Machine Learning [20]	This research explains how using IoT and machine learning can help predict crop diseases, which can save farmers money and boost crop yields.	Gathering current environmental data (temperature, humidity, and rainfall) with sensors in an IoT system. Analyzing the data with machine learning to predict crop diseases. Communicating disease predictions to farmers via text messages or a web browser.	The system might not always be accurate in predicting diseases. The sensors and algorithms could have limitations in accuracy and reliability.
4.	Nurturing Agribusiness: A Sustainable Farming System for Tomato Crop Management Leveraging Machine Learning [36]	The research combines IoT and machine learning to manage tomato crops more effectively, and it helps with detecting diseases and supports marketing efforts.	Use of TCS3200 color sensor to collect RGB values of tomatoes. Training a machine learning model (random forest) using RGB data. Utilizing Google Firebase for real-time communication and data storage between farmers and buyers. Integrating Micro ML to enable machine learning on the microcontroller.	Wear and tear sensitivity of the hardware units and the need for regular maintenance.
5.	IoT and Machine Learning System for Early/Late Blight Disease Severity Level Identification on Tomato Plants [37]	An IoT and machine learning system for early/late blight disease severity level identification on tomato plants.	The study developed an IoT and machine learning system with five key subsystems: control, data acquisition, data storage, machine learning, and data visualization. The machine learning component achieved a mAP of 77.25% with a 3.71 s average inference time.	The system is still under development and has not been fully implemented or evaluated. The model’s performance has not been compared to other models, leaving its real-world readiness unclear.
6.	Disease Detection in Tomato plants and Remote Monitoring of agricultural parameters [38]	The paper integrates IoT sensors and deep learning for remote monitoring of tomato plant health and agricultural parameters.	Detecting tomato leaf diseases using a convolutional neural network. Measuring soil moisture, temperature, and humidity with sensors. Delivering sensor data and disease detection results to farmers via a mobile app.	Potential variability in the CNN’s accuracy due to the quality of training data and possible calibration issues with sensors measuring soil moisture, temperature, and humidity.
7.	Cloud-based Tomato Plant Growth and Health Monitoring System using IoT [39]	An IoT-based system integrates soil monitoring and machine learning to predict and monitor tomato plant growth and health.	Soil moisture sensors control water relays. Humidity–temperature sensors monitor environmental conditions. Camera module detects tomato plant diseases. Azure Custom Vision Model enhances disease detection accuracy.ThingSpeak displays real-time data on air temperature, humidity, soil temperature, and moisture levels.	Potential inaccuracies from soil moisture sensors and environmental sensors, limited generalization of the Azure Custom Vision Model to all disease types, and possible issues with real-time data display on ThingSpeak.

**Table 3 sensors-24-06177-t003:** Various sensors used in this study and their operating ranges, data format, and threshold band.

Sensor Type	Current	Voltage	Data Format	Threshold Band
NPK Sensor	10–20 mA	3.3–5 V	Analog	Low, medium, and high nutrient levels
Temperature Sensor	0.5–10 mA	3–5 V	Analog	Temperature range (e.g., −40 °C to 125 °C)
Humidity Sensor	0.5–15 mA	2.5–5 V	Analog	Humidity range (e.g., 0–100% RH)
GPS Sensor (NEO-6 m)	20–100 mA	3–5 V	Digital (NMEA, etc.)	Geographical coordinates
Wi-Fi Sensor	15–200 mA	3.3–5 V	Digital (TCP/IP, etc.)	Signal strength (dBm)
RGB Color Sensor	10–30 mA	2.7–5.5 V	Digital (RGB values)	Color intensity range

**Table 4 sensors-24-06177-t004:** Data preparation stage.

Timestamp	NPK Level	Temperature (°C)	Humidity (%)	pH Value	Diseased
27-11-2023 10:00	2	22	45	6.7	1
27-11-2023 10:01	1	23	47	5.6	0
27-11-2023 10:02	3	22	50	6.7	0
27-11-2023 10:03	2	21	48	6.8	0
27-11-2023 10:04	1	22	46	5.5	0
27-11-2023 10:05	3	23	49	4.4	0
27-11-2023 10:06	2	24	45	6.7	1
27-11-2023 10:07	1	21	47	6.6	0
27-11-2023 10:08	3	22	50	6.7	0
27-11-2023 10:09	2	23	48	6.6	0

**Table 5 sensors-24-06177-t005:** A tabular summary of earlier ML models along with the present research results.

S. No	Reference and Number	Classification Method	Database and Disease Type	ML-IoT Enabled	Main Findings	Limitations	Accuracy Measure
**1.**	Kapucuoglu and Kirci (2021) [54]	CNN with hyperparameter optimization	New Plant Diseases Dataset—Kaggle and tomato leaf disease	No	Accuracy increased from 92% to 98% once a proper hyperparameter tuning was undertaken.	Only accuracy test was performed.	98%
**2.**	Kumar et al. (2024) [19]	Integration of machine learning and IoT	Plant Village database and early detection of tomato leaf disease	Yes	An integrated IoT and machine learning system for early detection of tomato leaf diseases.	The system lacks true real-time capabilities, requiring further research to achieve real-time functionality. The model has only been validated on benchmark datasets. Additional research is needed to assess its performance on real-world data.	Achieved accuracy of 95.6%
**3.**	Suchithra and Pai, (2020) [55]	Extreme learning machine (ELM)	Soil fertility indices—soil nutrient classification and improvement of dynamic soil parameters.	No	The ELM with Gaussian radial basis function excelled in classifying soil nutrient fertility indices. The ELM with hyperbolic tangent function was most effective for classifying soil pH levels. Parameter optimization was crucial for enhancing the ELM model’s performance in soil fertility classification.	The study could be expanded to classify additional soil nutrients beyond OC, P, K, and B—the proposed model could be applied to other agro-ecological regions.	The primary outcomes measured in the paper are the classification and prediction of soil fertility indices (for organic carbon, phosphorus, potassium, and boron) and soil pH levels.
**4.**	Suneja et al. (2022) [39]	IoT-based machine learning	Realistic sensors used and tomato leaf detection	Yes	An IoT-based system integrates soil monitoring and machine learning to predict and monitor tomato plant growth and health.	The sensors provide valuable data but do not capture all factors affecting plant health, limiting effectiveness. The ThingSpeak platform aids real-time monitoring, but relies on stable internet, which can be unreliable in remote areas.	Not provided
**5.**	Wagle et al., (2024) [56]	Bilinear LSTM with Bayesian Gaussian optimization	Weather data from local weather stations and tomato plant disease	No	Bilinear LSTM with Bayesian Gaussian optimization is used to predict tomato plant disease using meteorological parameters.	The model, BLSTM_bayOpt, is tailored for the Pune region, and its effectiveness in other regions with different climates is untested, limiting its generalizability. The model’s accuracy in predicting temperature and humidity is crucial for disease forecasting, but any inaccuracies in weather data could impair its predictions.	Coefficient of determination-93.2%
**6.**	Bhatia et al., (2021) [57]	A forecasting technique	Weather-based prediction model developed from empirical data and powdery mildew disease in tomato plants	No	The research creates weather-based prediction models using KNN, decision tree, and random forest algorithms to predict powdery mildew disease in tomato plants at early stages.		Not measured.
**7.**	Dhivyaa et al. (2023) [58]	An enhanced deep learning model	Visual geometric data and tomato leaf disease prediction	No	An enhanced deep learning model using Bayesian optimization and data augmentation techniques improves tomato leaf disease prediction.	Tomato leaf disease symptoms can vary widely. There is a shortage of high-quality data to train the model. Disease identification and classification methods are not standardized. It is challenging to factor in environmental conditions like temperature and humidity. The training data need to be more diverse, which can be improved using data augmentation techniques.	Not measured.
**8.**	Wang and Liu (2024) [59]	Enhanced bidirectional weighted feature pyramid network (IBiFPN)	A custom dataset (10,000 tomato disease images), captured in a real greenhouse environment and tomato disease	No	The model enhances detection accuracy by integrating Swin-DDETR’s self-attention mechanism, improving small target disease detection. The use of Meta-ACON in the backbone network amplifies the model’s ability to depict disease features.	The model’s performance in greenhouses can be improved by expanding real-world tomato disease samples and adding autonomous continuous learning. Further research is needed to detect early patterns of high-incidence diseases for timely detection and prevention.	Tomato Det model achieved a mean average precision of 92.3%
**9.**	This present study	Bayesian optimization with KNN	Real-time and high-resolution data and tomato crop disease	Yes	Bayesian optimization with KNN	Lack of diversified databases to use in the present study.	Accuracy 95% Precision 95% Recall 95% F1 score 94%

## Data Availability

Data are contained within this present article, as part of the Appendix A.

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
