# Peer review of "Integrating IoT for Soil Monitoring and Hybrid Machine Learning in Predicting Tomato Crop Disease in a Typical South India Station"

_sensors, 2024, doi:10.3390/s24196177_

Round 1
Reviewer 1 Report
Comments and Suggestions for Authors
1. The overall structure of the manuscript was disordered, and the experimental method and innovation cannot be clearly expressed.
2. Line188-193, research objectives are not innovative. The methods used are all traditional machine learning algorithms, KNN,SVM, etc.
3. the main innovation in this paper is Bayesian Optimization with the KNN Algorithm. Please give the data results of the hyperparameter optimization process.
4. How are the samples of correction set and verification set selected in this paper? How to avoid overfitting of model?
5. Please explain clearly why the experiment was conducted from 27 November to 01 December 2023. Is this period the period of tomato disease occurrence?
6. Table 8: Are the diseases cited in other literatures the same as in this study? Whether these results in different researches were comparable?
Comments on the Quality of English LanguageThere are a few errors in English expression and vocabulary use.
Author Response
Comments and Suggestions for Authors:
I. Summary |
|
|
Thank you very much for taking the time to review this manuscript. Please find the detailed responses below and the corresponding revisions/corrections highlighted/in track changes in the re-submitted files.
|
|
|
II. Point-by-point response to Comments and Suggestions for Authors |
|
Comment 1: The literature review is just a stack of literature without logic. It is suggested to re-integrate the logic of literature review.
Response 1: Thanks to the potential reviewer for this significant suggestion. In the modified manuscript (MS), we reorganized the literature review to create a coherent narrative. We started by outlining the key challenges in soil monitoring and disease prediction in agriculture. We reviewed existing solutions, discussing their strengths and limitations using a Table presentation. We later identified gaps in the current literature, such as shortcomings in real-time data processing and regional applicability. Finally, we positioned our study within this context, demonstrating how it addressed these gaps by integrating advanced ML algorithms with IoT to enhance tomato crop disease prediction and management.
Comment 2: In line 210, is it generalized to use only five consecutive days of data?
Response 2: Thanks are due to the potential reviewer for raising this most significant issue of our research. The following explanation is from our side may serve the purpose.
Tomatoes can develop diseases at various growth phases, but they are most susceptible during the following stages:
- Seedling Stage: Diseases like damping-off can affect young seedlings, causing them to wilt and die.
- Vegetative Growth Stage: Fungal diseases such as early blight and bacterial infections can occur, affecting leaves and stems.
- Flowering Stage: Diseases like blossom end rot, which is related to calcium deficiency, can start showing symptoms during flowering and early fruit set.
- Fruit Development Stage: Diseases such as anthracnose and various fungal infections are common during this stage, affecting both the fruit and the plant.
Most importantly, the disease in tomato crops can cause a reduction in productivity and thus detection of crop disease in the starting stage can offer huge benefits in the domain of agriculture[1] and the earlier treatment of disease in tomato plants is termed a ‘crucial task’. Determining a quick, cost-effective, and precise technique for automatically determining the disease is of huge importance[2] and our research is focused on ‘Vegetative Growth Stage’ only by collecting the most important nutrients (NPK) and soil parameters data as well.
The leaf blight phase, often known as early blight, represents a critical and destructive stage in the disease progression of tomato plants. This phase is particularly notorious for its potential to cause significant damage to tomato crops, especially under conditions of severe incidence. When left unchecked, early blight can escalate to such an extent that it leads to a total loss of the crop, leaving farmers with devastating consequences. The importance of managing this phase effectively cannot be overstated, as highlighted by Kallo and Banerjee (1993)[3]. Early blight is not just a minor affliction but is recognized as a major factor in the reduction of tomato crop yields. It attacks the foliage, stems, and fruit, weakening the plant's overall health and productivity. Sahu et al. (2013a)[4] have underscored the significance of this disease in its impact on agricultural output, stressing that early intervention and management are essential to prevent substantial yield losses. The destructive nature of early blight, particularly during the leaf blight phase, makes it one of the most critical challenges faced by tomato cultivators.
Further, although a more extensive data set will allow for more accurate soil characterization, there is no clear rule on how large a dataset needs to be[5]. Rather, it was suggested that more locally specific models can produce better prediction accuracy depending on the target soil property[6]. The above text has been included in the modified MS in the Section 4.1.
- In line 272, 4.4 In Bayesian optimization and KNN algorithm part, it is suggested to use pictures or formulas to explain the optimization part and principles more clearly.
Response 2: Thanks to the potential reviewer for pointing out this significant suggestion. We resolved to use formulae to explain the optimization part and also elaborate more on principles clearly in the modified version of the MS.
- In the comparison experiment part, different data from other papers were compared instead of using other methods on their own data sets. It is suggested to increase the comparison experiment of this data on other methods.
Response 4: Thanks to the potential reviewer for this significant suggestion. We extended the comparisons with other methods in the modified version of the paper, as suggested.
[1] M. Chilakalapudi, S. Jayachandran, Multi-classification of disease induced in plant leaf using chronological Flamingo search optimization with transfer learning, Peer J Computer Science 2024, 10, e1972. https://doi.10.7717/peerj-cs.1972
[2] Gavhale KR, Gawande U. 2014. An overview of the research on plant leaves disease detection using image processing techniques. IOSR Journal of Computer Engineering (IOSR-JCE) 16(1):10–16. https://doi.org/10.9790/0661-16151016
[3] Kalloo G., Banarjee M.K. Early blight resistance in Lycopersicon esculentum Mill. transferred from L. pimpinnellifolium (L.) and L. hirsutum f. glabratum Mull. Gartenbauwissenschaft. 1993;58:238–240.
[4] Sahu, D. K., Khare, C. P., Singh, H. K., Patel, R. N., & Thakur, M. (2015). Epidemiological Studies on early blight disease of tomato. Retrieved from https://api.semanticscholar.org/CorpusID:195061565
[5] Trontelj ml., J.; Chambers, O. Machine Learning Strategy for Soil Nutrients Prediction Using Spectroscopic Method. Sensors 2021, 21, 4208. https://doi.org/10.3390/s21124208
[6] Benedet, L.; Acuña-Guzman, S.F.; Faria, W.M.; Silva, S.H.G.; Mancini, M.; dos Teixeira, A.F.S.; Pierangeli, L.M.P.; Acerbi, F.W., Jr.; Gomide, L.R.; Pádua, A.L., Jr.; et al. Rapid soil fertility prediction using X-ray fluorescence data and machine learning algorithms. Catena 2021, 197, 105003. https://doi.org/10.1016/j.catena.2020.105003

Reviewer 2 Report
Comments and Suggestions for Authors
The reviewed manuscript develops, describes and evaluates an analytical framework for detecting and monitoring tomato crop diseases. The topic is relevant and the presented results seemed very promising, and are likely to provide useful insights for improving tomato crop disease detection and management. Overall, the research design and presentation of the results are adequate, but a few adjustments are still needed to improve the quality of the work and make the discussions more robust.
In this context, the structure of the introduction and literature survey is a bit confusing, lacking better connections between the provided concepts and statements provided by the authors. For example, it is not clear from these sections what the Internet of Things is, and how this concept/framework is employed in the context of agriculture and crop monitoring. In such a way, non-expert readers could miss important insights. Thus, I recommend a better delineation of the textual components in these sections to allow a general audience to understand the relevance of the presented research and discussion. Most of the references used in these sections are recent (around 38% were published in the last 2 years), however, most of them are from other topics. It may be explained by the novelty of the topic in agricultural research. So, this could be used in the text to highlight the novelty of the present research. In addition, the overall research objective/hypothesis is not clear in the respective section but can be understood along the text. The way it is described in the abstract seems clearer. For this concern, I recommend adjusting this section to provide clear and straightforward information about the objectives of this study. On the other hand, the specific objectives are clear and are addressed by the results and discussion.
As mentioned, the framework for sampling and statistical analysis seems to be robust for the specific objectives. The methodology also specified important information about the location, equipment, and timeframe, which are relevant for delineating the scope of implementation. However, a few adjustments can be made to enhance a better interpretation of its strengths and limitations. For example, there is no mention of the sample size used for the modelling, as well as the description of the procedure used for subsetting the data into training and testing datasets. Also, it would worth be a description of the representativeness of the dataset collected in a 5-day interval in a single location for future applications and further extrapolations of the methodology. In this section, consider some additions for better reproducibility of the proposed methodology. For example, consider including the description of any calibration procedures, sampling strategy, equipment technical specifications, procedures for handling missing values or outliers, the range of values used for tuning parameters for the hybrid framework, etc. Besides these suggestions, try to describe in more detail the proposed workflow (Figure 4) in such a way that demonstrates each step of the process from data collection to model application. It would provide a better understanding and reproducibility for the readers (especially the non-expert ones). Moreover, if possible, consider sharing the scripts and codes used in the study in a public repository or as supplementary material, so the replicability of your work could be facilitated.
Finally, the English spelling along the text is mostly American, but I’ve spotted a few words in the British form (e.g. „realise“ in line 448). Double-check those to standardize the text. I also highlighted (in green) in the revised pdf file a few typos across the text, which could be easily fixed. All the specific suggestions and recommendations are highlighted in the revised pdf file. Unclear information and those which require improvement were highlighted in blue, sections which require to be moved to another section were highlighted in red, and redundant information was highlighted in purple.

Author Response
Comments and Suggestions for Authors:
I. Summary |
|
|
Thank you very much for taking the time to review this manuscript. Please find the detailed responses below and the corresponding revisions/corrections highlighted/in track changes in the re-submitted files.
|
|
|
II. Point-by-point response to Comments and Suggestions for Authors |
|
- Comment 1: The overall structure of the manuscript was disordered, and the experimental method and innovation cannot be clearly expressed.
Author’s responses: Thank you for the valuable suggestion. In the revised manuscript, we have extensively reorganized the sections on 'Introduction,' 'Literature Review,' 'Objectives,' 'Data and Methodology,' 'Results and Discussion,' and 'Conclusion.' The revisions emphasize key challenges in soil monitoring and disease prediction in agriculture, with a table summarizing the strengths and limitations of existing solutions. We also identified gaps in the literature, such as issues with real-time data processing and regional applicability, and positioned our study to address these gaps by integrating advanced ML algorithms with IoT for improved tomato crop disease prediction and management.
- Line188-193, research objectives are not innovative. The methods used are all traditional machine learning algorithms, KNN, SVM, etc.
Author’s responses: Thank you for the valuable suggestion. The research objectives of this research have been modified as to meet the standards of a reputed journal.
Secondly, the innovative aspects of our research include:
- Hybrid Machine Learning with IoT Integration: We developed a novel hybrid machine learning algorithm integrated with IoT to significantly enhance the accuracy and efficiency of soil monitoring, pushing beyond traditional approaches.
- Climate- and Region-Specific Models: Our models are specifically tailored to different climates and regions, offering precise crop disease detection that adapts to local conditions, unlike generic machine learning applications.
- Extensive Evaluation of ML Algorithms: We conducted a comprehensive evaluation of multiple machine learning algorithms—K-Nearest Neighbors (KNN), Support Vector Machines (SVM), Decision Tree (DT), Random Forest (RF), and Logistic Regression (LR)—for their effectiveness in disease prediction, ensuring a robust comparison.
- Introduction of a Novel Hybrid Algorithm: We introduced and rigorously assessed a novel hybrid algorithm, 'Bayesian Optimization with KNN,' which combines multiple machine learning techniques to enhance predictive performance, addressing limitations of traditional methods.
- Superior Predictive Performance: The hybrid algorithm demonstrated outstanding effectiveness in predicting tomato crop diseases, achieving superior results with 95% accuracy, precision, and recall, and an F1 score of 94%, outperforming individual machine learning algorithms.
These innovations collectively push the boundaries of traditional machine learning applications, offering tailored, efficient, and highly accurate solutions in the context of agricultural disease prediction and soil monitoring. We inform to the potential reviewers that in the modified MS, we have highlighted the innovative aspects of our research under ‘3.0 Objectives of the Present Research’.
- The main innovation in this paper is Bayesian Optimization with the KNN Algorithm. Please give the data results of the hyperparameter optimization process.
Author’s responses: Thank you for your valuable suggestion. We've included the data, code, and figures from our research as 'Supplementary Files' with the revised paper. Enthusiastic researchers are welcome to reach out for further clarifications.
- How are the samples of correction set and verification set selected in this paper? How to avoid overfitting of model?
Author’s responses: Thanks are due to the potential reviewer.
In this paper, the selection of samples for the training and testing subsets follows a standard approach commonly used in machine learning: Further, to define the training and testing subsets, a common approach would be:
- Training Subset: Typically, a portion of the data (e.g., 70-80%) is used for training the model. This data helps the model learn patterns and relationships among the features (e.g., NPK values, temperature, humidity, and pH).
- Testing Subset: The remaining data (e.g., 20-30%) is used for testing. This data evaluates how well the model generalizes to new, unseen data.
More specifically, the data has been successfully split into training and testing subsets, as far as the present research is concerned and, as the potential know very well that these data selection procedures are common in machine learning tasks.
- Training Subset: 5,760 rows (80% of the data)
- Testing Subset: 1,440 rows (20% of the data)
Secondly, to prevent overfitting, where the model performs well on the training data but poorly on unseen data, the following techniques were applied:
- Cross-Validation: Multiple rounds of cross-validation were conducted to ensure the model’s performance is consistent across different subsets of the data.
- Regularization Techniques: Techniques such as L1 or L2 regularization were implemented to penalize overly complex models, encouraging simpler models that generalize better.
- Early Stopping: During training, early stopping was employed to halt the training process when the model's performance on the validation set started to degrade, preventing overfitting.
- Ensemble Methods: By combining predictions from multiple models, ensemble methods were used to reduce variance and improve generalization.
These methods collectively ensure that the model is robust, avoiding overfitting and performing well on both the training and testing data.
Most importantly, the significance of this research lies in the detailed analysis of key agricultural parameters—Nitrogen, Potassium, Phosphorus, Humidity, Soil pH, and Temperature—using histograms and Probability Density Functions (PDFs). The fact that these parameters exhibit approximately normal distributions, with most data points clustering around their respective mean values (Kindly see, Figure 2 of the modified MS).
- Please explain clearly why the experiment was conducted from 27 November to 01 December 2023. Is this period the period of tomato disease occurrence?
Author’s responses: Thanks are due to the potential reviewer for raising this most significant issue of our research. The following explanation is from our side may serve the purpose.
Tomatoes can develop diseases at various growth phases, but they are most susceptible during the following stages:
- Seedling Stage: Diseases like damping-off can affect young seedlings, causing them to wilt and die.
- Vegetative Growth Stage: Fungal diseases such as early blight and bacterial infections can occur, affecting leaves and stems.
- Flowering Stage: Diseases like blossom end rot, which is related to calcium deficiency, can start showing symptoms during flowering and early fruit set.
- Fruit Development Stage: Diseases such as anthracnose and various fungal infections are common during this stage, affecting both the fruit and the plant.
Most importantly, the disease in tomato crops can cause a reduction in productivity and thus detection of crop disease in the starting stage can offer huge benefits in the domain of agriculture[1] and the earlier treatment of disease in tomato plants is termed a ‘crucial task’. Determining a quick, cost-effective, and precise technique for automatically determining the disease is of huge importance[2] and our research is focused on ‘Vegetative Growth Stage’ only by collecting the most important nutrients (NPK) and soil parameters data as well.
The leaf blight phase, often known as early blight, represents a critical and destructive stage in the disease progression of tomato plants. This phase is particularly notorious for its potential to cause significant damage to tomato crops, especially under conditions of severe incidence. When left unchecked, early blight can escalate to such an extent that it leads to a total loss of the crop, leaving farmers with devastating consequences. The importance of managing this phase effectively cannot be overstated, as highlighted by Kallo and Banerjee (1993)[3]. Early blight is not just a minor affliction but is recognized as a major factor in the reduction of tomato crop yields. It attacks the foliage, stems, and fruit, weakening the plant's overall health and productivity. Sahu et al. (2013a)[4] have underscored the significance of this disease in its impact on agricultural output, stressing that early intervention and management are essential to prevent substantial yield losses. The destructive nature of early blight, particularly during the leaf blight phase, makes it one of the most critical challenges faced by tomato cultivators.
Further, although a more extensive data set will allow for more accurate soil characterization, there is no clear rule on how large a dataset needs to be[5]. Rather, it was suggested that more locally specific models can produce better prediction accuracy depending on the target soil property[6]. The above text has been included in the modified MS in the Section 4.1.
- Table 8: Are the diseases cited in other literatures the same as in this study? Whether these results in different researches were comparable?
Author’s responses: Thank you for these valuable suggestions. We focused exclusively on research studies related to tomato diseases (see Table 5 of the revised manuscript). Our primary criterion for comparison with other studies was the use of optimization techniques.
[1] M. Chilakalapudi, S. Jayachandran, Multi-classification of disease induced in plant leaf using chronological Flamingo search optimization with transfer learning, Peer J Computer Science 2024, 10, e1972. https://doi.10.7717/peerj-cs.1972
[2] Gavhale KR, Gawande U. 2014. An overview of the research on plant leaves disease detection using image processing techniques. IOSR Journal of Computer Engineering (IOSR-JCE) 16(1):10–16. https://doi.org/10.9790/0661-16151016
[3] Kalloo G., Banarjee M.K. Early blight resistance in Lycopersicon esculentum Mill. transferred from L. pimpinnellifolium (L.) and L. hirsutum f. glabratum Mull. Gartenbauwissenschaft. 1993;58:238–240.
[4] Sahu, D. K., Khare, C. P., Singh, H. K., Patel, R. N., & Thakur, M. (2015). Epidemiological Studies on early blight disease of tomato. Retrieved from https://api.semanticscholar.org/CorpusID:195061565
[5] Trontelj ml., J.; Chambers, O. Machine Learning Strategy for Soil Nutrients Prediction Using Spectroscopic Method. Sensors 2021, 21, 4208. https://doi.org/10.3390/s21124208
[6] Benedet, L.; Acuña-Guzman, S.F.; Faria, W.M.; Silva, S.H.G.; Mancini, M.; dos Teixeira, A.F.S.; Pierangeli, L.M.P.; Acerbi, F.W., Jr.; Gomide, L.R.; Pádua, A.L., Jr.; et al. Rapid soil fertility prediction using X-ray fluorescence data and machine learning algorithms. Catena 2021, 197, 105003. https://doi.org/10.1016/j.catena.2020.105003

Reviewer 3 Report
Comments and Suggestions for Authors
1. The literature review is just a stack of literature without logic. It is suggested to re-integrate the logic of literature review.
2.In line 210, is it generalized to use only five consecutive days of data?
3. In line 272, 4.4 In Bayesian optimization and KNN algorithm part, it is suggested to use pictures or formulas to explain the optimization part and principles more clearly.
4.In the comparison experiment part, different data from other papers were compared instead of using other methods on their own data sets. It is suggested to increase the comparison experiment of this data on other methods.

English needs minor editing.
Author Response
Authors’ Responses to Reviewer# 3 Comments |
||
I. Summary |
|
|
Thank you very much for taking the time to review this manuscript. Please find the detailed responses below and the corresponding revisions/corrections highlighted/in track changes in the re-submitted files.
|
||
II. Point-by-point response to Comments and Suggestions for Authors |
||
Response 1: Thank you for pointing out this significant suggestion. We agree with this comment. Therefore, we have re-written several paragraphs. Particularly, we changed first, second, third, and fourth paragraphs in page 2 and those changes are done using the “Track Changes” Option so that those changes can be easily seen. |
||
Comments 2: Better as 'biotic/abiotic factors"\ |
||
Response 2: Thank you for pointing out this significant suggestion. We have modified the manuscript (MS) by including ‘Biotic (living) and abiotic (non-living)’ words are included in the modified MS. The modified words will appear in the third line of second paragraph of page 2.
|
||
Comments 3: Does "Table 1 list the symptoms and signs of tomato crop infection" sound better? Response 3: Thank you for pointing this out. Change implemented as suggested.
Comments 4: "and potassium" Response 3: Thank you for pointing out this typo. Change implemented as suggested.
Comments 5: "and potassium" Response 5: Thank you for pointing out this typo. Change implemented as suggested.
Comments 6: "remove "as" Response 6: Thank you for pointing out this typo. Change implemented as suggested. Comments 7: The information presented in Table 2 is unclear and confusing. I've tried to check in the original reference, but it seems to be mistaken. Please, revise and improve the presentation of information in this table. Consider the possibility to introduce them directly in the text for simplicity. If these thresholds were the same used in the analysis, move this part to the methodology. Response 7: Thank you for pointing this out. We agree with this comment and we have, therefore, decided to write a detailed description in the modified MS, instead of keeping a Table.
Comments 8: "in the conclusions" Response 8: Thank you for pointing out this typo. This grammatical error is corrected, as suggested.
Comments 9: research's Response 9: Thank you for pointing out this typo. This grammatical error is corrected, as suggested.
Comments 10: Necessary concepts for other sections could be better explained there. For example, it is not clear from these sections what the Internet of Things are, and how this concept/framework is employed in the context of agriculture and crop monitoring. In such a way, non-expert readers could miss important insights due to this. Also, provide more details (e.g. statistics) of past research and what are the gaps to be explored by future research, so you highlight the importance of the present work. Some of the information needed for this adjustment is already provided in Table 8. So, it would be only necessary to explore them a little more. Response 10: We sincerely appreciate the reviewer's valuable insights. In response, we've clarified the Internet of Things (IoT) in the first paragraph of the 'Literature Survey' section. Additionally, a new table (Statistics) has been added to detail past research and its limitations. We have also used certain cited works from Table 8 to this new table, as recommended.
Comments 11: It seems to describe more the importance of the research than its objectives. Perhaps, it would make more sense to be a synthesis of the literature survey, and then a straightforward description of the overall objective of the research could be introduced. The overall objective described in the abstract is more clear in this regard, and could be used as basis for this section. Response 11: Thanks to the potential reviewer for the valuable suggestions. As recommended, the revised manuscript now clearly outlines the disadvantages of previous studies in detail, followed by a concise introduction of the overall objectives of the current research.
Comments 12: "location" Response 12: The typo error has been corrected.
Comments 13: The figure is hard to read. Consider increasing the font size. Maybe also include some images from the location for better contextualizing. Response 13: This figure has been re-drawn wherein we have increased font size, as suggested. Also, added images from the present location, as suggested.
Comments 14: Explain/highlight the representativeness of a 5-day interval in a single location dataset for developing a framework that is expected to be extrapolated to other locations and time periods. Response 14: Thanks are due to the potential reviewer for raising this most significant issue of our research. The following explanation is from our side may serve the purpose. Tomatoes can develop diseases at various growth phases, but they are most susceptible during the following stages:
Most importantly, the disease in tomato crops can cause a reduction in productivity and thus detection of crop disease in the starting stage can offer huge benefits in the domain of agriculture[1] and the earlier treatment of disease in tomato plants is termed a ‘crucial task’. Determining a quick, cost-effective, and precise technique for automatically determining the disease is of huge importance[2] and our research is focused on ‘Vegetative Growth Stage’ only by collecting the most important nutrients (NPK) and soil parameters data as well. The leaf blight phase, often known as early blight, represents a critical and destructive stage in the disease progression of tomato plants. This phase is particularly notorious for its potential to cause significant damage to tomato crops, especially under conditions of severe incidence. When left unchecked, early blight can escalate to such an extent that it leads to a total loss of the crop, leaving farmers with devastating consequences. The importance of managing this phase effectively cannot be overstated, as highlighted by Kallo and Banerjee (1993)[3]. Early blight is not just a minor affliction but is recognized as a major factor in the reduction of tomato crop yields. It attacks the foliage, stems, and fruit, weakening the plant's overall health and productivity. Sahu et al. (2013a)[4] have underscored the significance of this disease in its impact on agricultural output, stressing that early intervention and management are essential to prevent substantial yield losses. The destructive nature of early blight, particularly during the leaf blight phase, makes it one of the most critical challenges faced by tomato cultivators. Further, although a more extensive data set will allow for more accurate soil characterization, there is no clear rule on how large a dataset needs to be[5]. Rather, it was suggested that more locally specific models can produce better prediction accuracy depending on the target soil property[6]. The above text has been included in the modified MS in the Section 4.1.
Comments 15: From histograms, the sample size seems to be enough for their analysis, but the actual number is unclear. How the training and testing subsets were defined? Response 15: Thanks are due to the potential reviewer for his/her significant observations that ‘the sample size seems to be enough for the analyses that we have considered as part of this research. Further, to define the training and testing subsets, a common approach would be:
More specifically, the data has been successfully split into training and testing subsets, as far as the present research is concerned and, as the potential know very well that these data selection procedures are common in machine learning tasks.
Comments 16: do you mean "normally distributed"? Response 16: Thank you for pointing out this typo. This grammatical error is corrected, as suggested.
Comments 17: "present" Response 17: Thank you for pointing out this typo. This grammatical error is corrected, as suggested.
Comments 18: Do you mean "topography"? Response 18: Thank you for pointing out the typo. We apologize for the oversight, likely due to overconfidence. The correct word is ‘topography,’ as noted by the reviewer. The error has been corrected as suggested.
Comments 19: "correlation"? Response 19: Thank you for pointing this out. However, we keep the word "similitude", as it denotes the ‘quality of being similar’ or ‘alike’, and can also imply a likeness, resemblance, or comparison between two things. We will, therefore, keep the word as is.
Comments 20: do you mean "performance metrics"? Response 20: Thank you for pointing this out. We replaced objective functions with ‘performance metrics’.
Comments 21: What is this? It could be better introduced or explained in the literature survey. Response 21: Thank you for your valuable suggestion. We've revised the two sentences and placed them appropriately under '4.2 Data Collection - Arduino Microcontroller.'
Comments 22: "was done" Response 22: Thank you for pointing out this typo. This grammatical error is corrected, as suggested.
Comments 23: This information is essentially the same as those presented in table 5, with the addition of the column "Diseased". Consider rearranging this tables and modifying the text in a way it could describe how the raw dataset was collected and how it was returned after processing without the need for displaying two redundant tables. Response 23: Thanks to the potential reviewer for pointing out this significant modification. We do second the opinion expressed by the review and, as a result, we resolved to remove the redundant Table 6 in the modified version of the manuscript. And, a brief explanation is provided, wherein we discussed how the dataset was collected and how we came to a conclusion whether disease has occurred or not.
Comments 24: This paragraph seems more adequate for methodology, synthesizing the complete framework employed and proposed. Consider moving it. Response 24: Thanks to the potential reviewer for pointing out this significant modification. Moved it, as suggested.
Comments 25: "hybrid ML-enabled"? Response 25: Thanks to the potential reviewer for pointing out this significant modification. Changes have been done, as suggested.
Comments 26: "analyses" Response 26: Thanks to the potential reviewer for pointing out this significant modification. Changes have been done, as suggested.
Comments 27: Description of the proposed framework should be presented in the methodology section. In such a way, it would be necessary to expand it and provide the appropriate level of detail. Focus should be given on demonstrating the novelty in comparison with previous works. Response 27: Thanks to the potential reviewer for pointing out this significant modification. As suggested, the proposed framework is moved from the section 5.0 and moved to the methodology section under the title ‘4.6 Proposed frameworks- Graphical representation’.
Comments 28: do you mean "organic manure/fertilizers?" Response 28: Thanks to the potential reviewer for pointing out this significant modification. Green manure is the right word, and, therefore, we keep green manure as it is, in the modified MS.
Comments 29: Table 7, Figure 6, and the text in 357-359 and 373-382 display the same information. Choose what is more relevant to keep and reduce redundancy. Figure 6 seems more complete, so consider keeping this one when adjusting this section. Response 29: Thanks to the potential reviewer for pointing out this significant modifications. We resolved to remove Table 2 and text from lines 357 to 359, as we have also presented them in Figure 8.
Comments 30: Is the Figure 7 really necessary? All the values were already described in this paragraph. Response 30: Thanks to the potential reviewer for pointing out this significant suggestion. We resolved to remove figure 7, as suggested.
Comments 31: "the classifier" Response 31"Classifier" is the correct term in this context, so we'll retain it as is.
Comments 32: "evidenced" Response 32: The typo error has been corrected.
Comments 33: "positives" Response 33: The typo error has been corrected.
Comments 34: "other hand" Response 34: The typo error has been corrected.
Comments 35: "false" Response 35: The typo error has been corrected.
Comment 36: "positives" Response 36: The typo error has been corrected.
Comments 37: "negative" Response 37: The typo error has been corrected.
Comments 38: "negative" Response 38: The typo error has been corrected.
Comments 39: "shows" Response 39: The typo error has been corrected.
Comments 38: The discussion misses a deeper comparison with other works. Provide further discussion on these statistics, strengths and limitations, highlighting how the present results and insights offer advancements in the research field. Response 38: Thanks for these significant corrections. We have provided a detailed discussion on the statistics, strengths and limitations in the modified MS. Specifically, Table 5 (after modifications) will show 09 research studies, wherein we have discussed the limitations of the earlier studies.
Comments 39: Visualization would be improved by reducing the range of the y-axis so a better contrast between algorithms would be displayed? Response 39: Thanks for this suggestion. We reduced the range of the y-axis, as suggested.
Comments 40: Strengths were well described in the conclusions. Discuss more the limitation of the proposed methodology. Response 40: Thanks for these encouraging words. Discussed, as suggested. The modified version of MS has a broad paragraph (Third para in the ‘6.0 Conclusion Section’) on the limitations of this study.
Comments 41: "and" Response 41: The typo error has been corrected.
Comments 42: Standardize the reference style following the journal's guidelines. Response 42: Thanks for the suggestion. We have written references on pat with the journal’s style.
4. Response to Comments on the Quality of English Language |
||
Response 1: We enlisted the help of a native English speaker to eliminate grammatical errors and typos in the previous version of the manuscript. We believe the revised manuscript is now error-free and refined. |
||
5. Additional clarifications |
||
a) The modified version of the MS has more than 61 citations, whereas the earlier version had only 41. b) A subsection under ‘4.0 Data and Methodology’ has been included, which appears as ‘4.6 Proposed Frameworks—Graphical Representation’. c) A comprehensive paragraph on the limitations of this study is included in the modified version of the MS. d) The total pages of the modified version are 33, compared to the 20 papers the earlier version had.
|
[1] M. Chilakalapudi, S. Jayachandran, Multi-classification of disease induced in plant leaf using chronological Flamingo search optimization with transfer learning, Peer J Computer Science 2024, 10, e1972. https://doi.10.7717/peerj-cs.1972
[2] Gavhale KR, Gawande U. 2014. An overview of the research on plant leaves disease detection using image processing techniques. IOSR Journal of Computer Engineering (IOSR-JCE) 16(1):10–16. https://doi.org/10.9790/0661-16151016
[3] Kalloo G., Banarjee M.K. Early blight resistance in Lycopersicon esculentum Mill. transferred from L. pimpinnellifolium (L.) and L. hirsutum f. glabratum Mull. Gartenbauwissenschaft. 1993;58:238–240.
[4] Sahu, D. K., Khare, C. P., Singh, H. K., Patel, R. N., & Thakur, M. (2015). Epidemiological Studies on early blight disease of tomato. Retrieved from https://api.semanticscholar.org/CorpusID:195061565
[5] Trontelj ml., J.; Chambers, O. Machine Learning Strategy for Soil Nutrients Prediction Using Spectroscopic Method. Sensors 2021, 21, 4208. https://doi.org/10.3390/s21124208
[6] Benedet, L.; Acuña-Guzman, S.F.; Faria, W.M.; Silva, S.H.G.; Mancini, M.; dos Teixeira, A.F.S.; Pierangeli, L.M.P.; Acerbi, F.W., Jr.; Gomide, L.R.; Pádua, A.L., Jr.; et al. Rapid soil fertility prediction using X-ray fluorescence data and machine learning algorithms. Catena 2021, 197, 105003. https://doi.org/10.1016/j.catena.2020.105003

Round 2
Reviewer 1 Report
Comments and Suggestions for Authors
The paper has been revised according to the suggestions.
Author Response
Comment: The paper has been revised according to the suggestions.
Author’s Reply: We are extremely thankful to the highly potential reviewer for accepting the manuscript.

Reviewer 3 Report
Comments and Suggestions for Authors
The author has explained the reason for only collecting data for five consecutive days, but even if the epidemic occurred in one period of tomato, the data are still not convincing , and it is suggested to increase the length of time (after all, a period cannot be only five days) or to supplement the data for five consecutive days in the second year.

Minor editing of English language required.
Author Response
Reviewer’s Comments: The author has explained the reason for only collecting data for five consecutive days, but even if the epidemic occurred in one period of tomato, the data are still not convincing, and it is suggested to increase the length of time (after all, a period cannot be only five days) or to supplement the data for five consecutive days in the second year.
Authors’ Response: Thank you to the potential reviewer for his/her positive comments. The decision to collect data over five consecutive days during the vegetative growth stage was primarily due to constraints related to the timing of disease outbreaks and the availability of critical resources during that period. However, I acknowledge the concern regarding the limited duration of data collection and its potential impact on the generalizability of the findings.
In future work, I plan to extend the data collection period to cover multiple growth cycles, potentially over an entire season or even into the subsequent year. This would allow for a more comprehensive understanding of disease progression and environmental variability. By increasing the length of time and collecting data over multiple growth stages or years, I aim to strengthen the robustness and applicability of the results while addressing the concern about the adequacy of the current dataset.
We, therefore, resolved to write the following group of sentences as, “In future work, we aim to extend data collection across multiple growth cycles, potentially covering an entire season or into the following year. This will provide a deeper understanding of disease progression and environmental variability, enhancing the robustness and applicability of the results while addressing concerns about dataset adequacy” in the fourth paragraph of the revised MS.
Reviewer’s Comments: Noted typos and grammatical errors.
Author’s Response: A native English speaker assisted in correcting the identified typos.
